# Post-Traumatic Play in Child Victims of Adverse Childhood Experiences: A Pilot Study with the MCAST—Manchester Child Attachment Story Task and the Coding of PTCP Markers

**DOI:** 10.3390/children9121991

**Published:** 2022-12-18

**Authors:** Daniela D’Elia, Luna Carpinelli, Giulia Savarese

**Affiliations:** Department of Medicine, Surgery and Dentistry “Scuola Medica Salernitana”, University of Salerno, 84081 Baronissi, Italy

**Keywords:** ACE, trauma, MCAST, post-traumatic play, attachment

## Abstract

**Background:** Play is among the most frequently observed distorted behaviors in victims of Adverse Childhood Experiences (ACEs). Terr’s (1981) studies helped to describe this behavioral distortion as post-traumatic child’s play (PTCP). This study aimed to evaluate whether child victims of Adverse Childhood Experiences (ACEs) engaging in semi-structured play present the markers of post-traumatic child’s play (PTCP) during the administration of the Manchester Child Attachment Story Task (MCAST), whose playful stories activate the attachment system. **Methods:** The sample comprised 17 child victims of ACEs (mean age = 6.76). Children were evaluated using the Manchester Child Attachment Story Task. **Results:** The analysis of the play clearly revealed the presence of markers associated with the characteristics of the post-traumatic play described by Terr (1981), namely, repetition, revisiting, hyperarousal, and danger. In particular, the intrusiveness dimension was observed, which can be identified in the post-traumatic play by the presence of recurrent memories, dreams, and dissociative symptoms. **Conclusions:** Post-traumatic play is characterized by repetition, containing aspects, scenes, or sequences of the traumatic event, expressed explicitly or symbolically.

## 1. Introduction

Adverse Childhood Experiences (ACEs) are a set of adverse situations of an interpersonal nature that negatively impact the ideal developmental path of the child. This type of experience is often due to a significant lack of protection, physical and emotional, and by attachment figures, producing traumatic effects [1]. The personal identity of an individual is the product emerging dynamic of interpersonal relationships of each person and it is formed through social activity and experience [2]. Exposure to ACEs can determine, as a developmental outcome, the configuration of various psychopathological disorders through continuous interactions with a neglectful and mistreating attachment figure [3].

Play is a form of communication for all children. During the initial stages of therapy with a child who has suffered trauma, the access and exploration of disturbing material should be well-calibrated to minimize the child’s emotional and behavioral dysregulation and maximize stabilization. Play therapy’s best practice focuses on safety [4], intersubjectivity [5], the social engagement system [6], play, and joy [7]. Through the observation of play sessions, it is possible to detect the child’s personal way of dealing with the new situations presented, his way of keeping in mind the primary relationships of attachment, the meanings he attributes to himself, to others, and the world. Attempts to deal with traumatic events in a child’s life the, through strictly verbal processing, can leave layers of the traumatic experience intact and infesting. Play activities can be used both as an emotional state change strategy and as occasions for experimentation with positive effects, regulation, and connection, using some of the following general criteria as a specific indication: (a) the child has suffered early trauma; (b) the child has attachment traumas, it is not possible for him to feel safe, and he shows poor relational resources; (c) the child is unable to explore traumatic memories and dies or becomes hyperactivated; and (d) the child shows an impairment of the social engagement system [8]. 

Over the years, direct observation and analysis of post-traumatic play have highlighted some important differences compared to the typical play of a child who has suffered or has not suffered trauma. Ordinary play is spontaneous; the playful stories evoked by the child are characterized by a story in which the events follow one another in history in their narrative succession. It is, therefore, usual to recognize an initial part, a central part, and an end. Most of the time they end with a “happy ending”; it stimulates the repair processes which is fundamental for the emotional well-being of the child. Conversely, a child’s post-traumatic play is ritualistic, repetitive, and often characterized by a nefarious ending. Those who practice ordinary play do so for fun and entertainment purposes, while post-traumatic play is used as a means to relieve the intense anxiety produced by the traumatic event to which the victim child was exposed and to repeat some parts of the trauma itself, with the main purpose of processing that event. Two different types of post-traumatic play can be distinguished: the positive type and the negative type [9]. In psychotherapy sessions, it is observed how in the “positive” post-traumatic play, the child reconstructs the traumatic event by modifying the negative elements of the trauma with the help of the therapist. Furthermore, the child feels serene and in control of his/her fantasy world during play psychotherapy sessions. In “negative” post-traumatic play, the repetition of play provides no relief from anxiety, and the child is unable to process or accept the trauma. During play sessions, these children are often anxious and do not have total control of their fantasies, and their repetition does not alleviate their internal conflicts. Consequently, “negative” post-traumatic play can be particularly dangerous, as it can worsen the effects of the trauma and lead to further regression. 

## 2. ACEs and Characteristics of Post-Traumatic Play of the Children 

The inability to modulate painful emotions and the amount of suffering experienced appear to be the most widespread effects of victimization on the child. He attempts to reduce internal tension and suffering [10] through abnormal and dysfunctional behavioral responses, thereby reinforcing ineffective mechanisms that only temporarily relieve traumatic stress. 

In her important clinical study [11], Terr defines play as an activity that the child *“feels he or she enjoys, either alone or in a group”* (p. 742).

Play is one of the most frequently observed distorted behaviors in young victims of ACEs. Terr describes this behavioral distortion as post-traumatic child's play (PTCP), which differs in meaning and function from the play behavior of non-victim children. Terr identified these PTCP characteristics.

-Compulsive repetitiveness. Children who have been traumatized play their “game” repeatedly until they are invited to stop or until they reach an emotional understanding of the connection of their game with the original psychic trauma.-An unconscious link between post-traumatic play and the traumatic event.-The literalness of play accompanied by simple defenses. In post-traumatic play, there is a simple repetition of the experience or simple defensive processing, such as identification with the aggressor.-The failure of play to relieve anxiety. Unlike ordinary play, which relieves anxiety, post-traumatic play provides no relief.-The wide age range of “players”. It been found that neither fully pubescent adolescents, nor 15- to 16-month-old infants, spontaneously play unique or personalized imaginative games.-The varying lag time prior to the development of post-traumatic play. It was found that some child victims developed post-traumatic play some months later, while others very soon after the traumatic event occurred. This post-traumatic time skew is connected with the sense of warning and impending disaster, which a child develops after a psychic trauma.-The “contagion” of post-traumatic play in new generations of children. Post-traumatic play can involve both non-traumatized and traumatically anxious children.-The lack of a happy ending. In post-traumatic play, there is no possibility of a happy ending after psychic trauma, as the child victim already knows what happened and it was not pleasant for him/her.-Revisiting. The possibility of tracing post-traumatic play to earlier trauma in that it is possible to bring out the traumatic traces of early trauma through play in order to reshape the states of distress related to it. This characteristic was redefined by Van der Hart [12], as a re-experience of the original trauma through specific movements, sensations, auditory and visual perceptions, emotional states, and ideas related to the event itself. Therapeutic action must continue to support passages in which it is quite common for other traumatic memories and other parts of the personality to emerge. This can occur particularly in tertiary structural dissociation. This return to traumatic memories of trauma is an expected development of treatment in complex cases of traumatization and, in these cases, there is a temporary revisiting.-The danger of post-traumatic play. The intensity of play and the need to share distress are so compelling that many children indulge in this need, regardless of the risks.

Therefore, an observation of the post-traumatic play reveals the failure of the symbolic distancing inherent in the nature of the playful experience and that the young victims identify only with themselves, without being able to distance themselves from the unfavorable event or to adopt other types of defenses. Moreover, the ending always appears catastrophic or is absent, reflecting an inability to find a solution to the traumas suffered.

Consequently, if the trauma to which the child was exposed in the real world has not yet been processed, something similar to the adverse experience can occur during the game on a symbolic level; it is possible, therefore, that the child experiences trauma through the intrusion of mental images in the form of beliefs or emotions that occur in the form of a disproportionate reaction to the current situation. This happens, for example, in the stories evoked by the Manchester Child Attachment Story Task - MCAST [13] that activate the begging system by eliciting emotions of fear and danger, such as the “Hurt knee” in which the child is told that the doll protagonist of the game has hurt her knee while he played in the garden and the child is asked to continue the narration. It has often been observed that non-traumatized children continue the story by recognizing the adult a role of care and comfort (e.g., *“the mother hears the child cry, runs to him, hugs him and heals his wound…”*), continuing the ordinary game freely. In the post-traumatic game, on the other hand, the response to the incipit of this story creates further suffering (e.g., *“The child stays in the garden with his knee bleeding, he is afraid of being seen by his mother, the mother then hears him, he arrives and he hits him because he shouldn't have hurt himself, he's stupid and he hits him so he learns!”*).

In post-traumatic play, repetition of play provides no relief from anxiety and the child is unable to process or accept the trauma. During play sessions these children are often anxious, they do not have total control of their fantasies, and their repetition does not alleviate their internal conflicts [8,9,11,14,15,16,17].

Brown [18] says that *“the purpose of play is to act […] to creativity, which might be seen as the source of all mental health*” (p. 11). This is a means of healing trauma, neurosis, and psychic illness. Humble et al. [19] performed a systematic review that examines the literature on the efficacy of child-centered play therapy for young people who have experienced traumatic events. The results related to internalizing problems, self-concept, and self-competence were somewhat consistent The Association for Play Therapy Board of Directors [20] affirmed that play therapy is always a developmentally congruent intervention that helps children enhance their capacity for self-regulation, build coping skills, and promote positive self-esteem. 

In 2018, Schaefer and Drewes [21] examined several mechanisms through which play can alleviate fears and anxieties. For example, play can evoke a positive effect and reduce stress, which counteracts negative effects. Furthermore, the increase in divergent thinking and creativity is often associated with play; the child is always looking for new combinations and discoveries in play that will then lead him to have more resources to solve personal and social problems.

On these premises, therefore, the objective of this study was to observe whether the playful stories of the MCAST [13], which evoke patterns of behavior and response by the child based on internal operational models—mental representations—of attachment relationships, reveal the markers of post-traumatic play (PTCP) that have developed in child victims of ACEs.

## 3. Materials and Methods

### 3.1. Participants

The reference sample of this descriptive study was composed of seventeen children who are victims of ACEs (nine ales, eight females, mean age = 6.76, and SD = 0.831). The children were taken in for a psychodiagnostic evaluation at institutions and specialized services in the diagnosis and treatment of neurodevelopmental disorders following a referral by public welfare services (41.2%), private services affiliated with the National Health System (35.3%), and private specialists (23.5%). The counseling services requested were the evaluation of the minors at a diagnostic level and the undertaking of a possible psychotherapy path (35.3%), the sole prescription of a psychological path (23.5%), and a psychodiagnosis at the Maternal-Child Service (41.2%).

Among the participants, 47.1% lived in their own home with their family of origin, 23.5% resided in a community, 17.6% were in foster care, and 11.8% had been adopted. In addition, 11.8% were victims of direct ACEs, 29.4% experienced indirect ACEs, and the remaining 58.8% were exposed to both types.

### 3.2. Procedure

The assessment process was carried out in 4–5 meetings in which, in addition to clinical interviews, the genogram, and the observation of the play, specific tests were administered, such as graphic and projective tests [22,23], the CBCL—Child Behavior Check List [24], and the CDI—Children’s Depression Inventory [25] for the evaluation of the emotional dimension. The MCAST—Manchester Child Attachment Story Task [13] was used for the assessment of attachment. The TSCC—Trauma Symptom Checklist for Children [26], the TSCYC—Trauma Symptom Checklist for Young Children—[27]; and the CSBI—Child Sexual Behavior Inventory—[28] were used to evaluate post-traumatic functioning.

In this study, according to the aims, we discussed only the results of the Manchester Child Attachment Story Task [13], a semi-structured assessment of play (for children between 4 and 8 years old). It is based on the process of completing stories within a controlled and repeatable setting, with the aim of evoking patterns of behavior and response by the child based on internal operational models—mental representations—of the attachment relationships that the child has developed. The tool is composed of six stories, whose protagonists are a doll child and a doll caregiver, structured as follows: an initial story, which has the aim of introducing the child to the methodology; four central stories, which present circumstances capable of eliciting the attachment system; and a final story, which aims to deactivate the attachment system and leave the child in a state of emotional serenity. After the imagined playful situation, the examiner asks the child about the mental states of the protagonists. As a result (it is a peculiarity of the interview), the child is continuously engaged, both emotionally and cognitively, in a situation capable of activating the attachment system through self-identification with the doll child, which, as per administration of the instrument, is named after the child who plays the game.

The administration of the MCAST required only one videotaped meeting with the prior consent of the caregiver. The coding system, with an inter-coder evaluation procedure, is based on the revision of the video recording and involves the use of scales relating to the following four fundamental dimensions: 1. the attachment behavior represented during the play; 2. consistency in the narrative; 3. the presence of disorganization phenomena; and 4. the capacity for mentalization and metacognition [13,29]. The coding is specific for each scale (in turn structured into different subscales) and the main measurements used in the research are (a) the strategy used/attachment classification; (b) the D score; and (c) derived codes, such as Safe/Insecure and D/not D.

As previously described, the intent of the MCAST is to evoke, within a controlled and repeatable setting, patterns of behavior and response on the part of the child based on internal operational models of attachment relationships. In this regard, let us analyze the first of the four stories entitled “Nightmare”, in which the child protagonist wakes up in the night terrified by a bad nightmare. Asking the child to continue the story, for example, we might have four different answers (each one explaining a different attachment pattern):The child is “Secure” could continue the story by saying that “*the little one cries, the mother immediately hears him and runs into his room, hugs him and reassures him, tells him that he will be scared by the bad alone, stays close to him and tells him a story to make him go back to sleep. The child calms down and falls asleep*”.Upon awakening from the nightmare, the “Avoidant” child will tend to self-care, e.g., “*The baby-doll wakes up, they say it's just a bad dream, the mother doesn't hear it, he turns over to the other side of the bed and waits to go back to sleep*”.The child with the “Anxious-Ambivalent” pattern, as we saw in the children in our study, might narrate that “*The baby doll wakes up terrified, screams, calls her mother, she arrives but cannot find him, he is hiding, the doll-mother yells at him not to scream and not to hide. She finds him, she tells him it's her fault that she doesn't have to watch video games until late so she has bad dreams! She scolds him and the baby doll continues to cry until she falls asleep*”.In the “Disorganized-disoriented” dyad, the child may narrate that the doll-child does not initially ask for help; when instead he asks for help, he says that the doll-mom arrives and beats him for waking her up, offends him by minimizing and not validating his fear, and there is no happy ending; for instance “…*he sends him to sleep down alone as a punishment!*”.

### 3.3. Data Analysis 

Data were analyzed with the IBM SPSS v.22 statistical program (IBM Corp, Armonk, NY, USA). ACEs were classified according to the method of Felitti et al. [30,31], which the DSM-5 (Criterion A) recognizes as inclusion criteria for PTSD—Post-Traumatic Stress Disorder [32].

In this contribution, we will present the results that emerged from the MCAST and from the observation of play, according to the markers described by Terr [11], for post-traumatic play.

For the dimensions examined, the coding system present in Green et al. [13] was used. The analysis of the play was carried out through the coding of the presence/absence of the markers associated with the characteristics of post-traumatic play (PTCP) described by Terr [11]: repetition, revisiting, hyperarousal, dangerousness, and happy ending.

## 4. Results

The frequencies of the types of ACEs (Table 1) were determined as follows: n = 4 children suffered sexual abuse, n = 4 experienced physical maltreatment, n = 2 experienced psychological maltreatment, n = 5 experienced neglect, and n = 2 were exposed to mental illness of the parent. All had experienced ACEs related to parental addictions to substances (alcohol and drugs). 

Through the semi-structured evaluation of the MCAST play, it was possible to observe the type of representation of attachment in children victims of ACEs. Table 2 shows that n° 10 children show the type D (Disorganized-disoriented), n° 2 the type B (Secure), n° 2 the type A (Avoidant), n° 3 the type C (Anxious-Ambivalent).

The frequencies of PTCP characteristics were analyzed to test whether MCAST stories activate markers of post-traumatic play in children victims of ACEs. Table 3 reveals a high frequency of scores for the repetition, revisiting, hyperarousal, and danger characteristics, while the variable happy ending is absent (or minimal).

By way of example, we must specify the characteristics of the various markers using the narrative developed by a child who participated in our study and who was the victim of physical abuse. In Story 1 “Hurt knee”, when the examiner asks him to continue the story with the doll characters starting from the skinned knee of the doll-child, the child begins to carry out an obsessive sequence, repeating repeatedly and compulsively the same series of behaviors and/or rituals (Repetition, 1): *“the doll-mother slaps the doll-child who has hurt her knee and puts him in the corner… after he goes back to the living room, but she still he slaps him and puts him in the corner, when he comes back later, he slaps him again and puts him in the corner…”*. 

In telling the story, the child initially maintains the dialogue in the third person, as required by the game, then changes to the first person and adds *“I really did sometimes deserve it!”*. In the meantime, we observe how the little one is agitated. We can see an increase in his level of activation (Hyperarousal, 3) by how he moves the doll characters (with force and impetus) and by his non-verbal behavior (general motor agitation). When he adds, continuing the story *“then in the end the child again comes out of the corner if he runs away from home…”*, it is highly likely that he is reliving the story (Revisiting, 2). In fact, we know from his medical history that after an episode of domestic violence and physical abuse, he ran away from home only to be found by the police and removed (which is why, among other things, he comes into consultation). His story, like most of those of traumatized children, will not end with a happy ending (Happy Ending, 5), stating *“he runs away from home, finds a ravine on the street, falls and dies”*; thus ending the story in a climate of general danger (Dangerousness, 4).

## 5. Discussion and Conclusions

The focus of the present research is on the MCAST, an assessment instrument that aims to evoke, in a controlled and repeatable context, patterns of behavior and response on the part of the child, based on internal operational models - mental representations - of attachment relationships that the child has developed.

The administration of the MCAST with the PTCP grid seems to be suitable as a structured collection strategy for such information. In fact, with reference to the age of our group, the intrusiveness dimension can be observed in the post-traumatic play by the presence of recurrent memories, dreams, and dissociative symptoms. Post-traumatic play is characterized by repetition, containing aspects, scenes, or sequences of the traumatic event, expressed explicitly or symbolically [33]. Attempts to process the experience by the infantile brain are expressed through the re-enactment of the events, with the compulsive reproduction of some aspects of the traumatic situation. The elaboration of the traumatic memory, on the other hand, is expressed through a symbolic exploratory play or through a play that reproduces the same themes of the traumatic event, but in an adaptive way; i.e., the contents are dynamically modified towards alternative endings, with the achievement of a state of calm and a return to exploratory free play [11].

Our results show that the ludic process of the MCAST made it possible to highlight another fundamental characteristic of psychic trauma through the indices extrapolated from the traumatic play. The process of completing the stories, in fact, made it possible to detect traumatic “repetition” responses to the event, which led to the intrusion of elements connected with the traumatic event in projections of the future [34,35,36]. Therefore, the symbolic function of play seems to be absent, where the child is unable to draw on the exploratory activity typical of the playful experience but instead uses play to re-enact blocked aspects of the trauma embedded in the memory networks in a dysfunctional way, enough to permeate the mental space and, therefore, also play behaviors [2,13,37,38]. At these ages, the intrusiveness dimension can be identified in post-traumatic play by the presence of recurring memories, dreams, and dissociative symptoms.

Post-traumatic play is characterized by repetition, containing aspects, scenes, or sequences of the traumatic event either explicitly expressed or symbolically represented. Attempts to process the experience by the infantile brain are expressed through the re-enactment of the events, with the compulsive reproduction of some aspects of the traumatic situation, but this can also take place through drawing. Post-traumatic play contains many elements of reality, to the detriment of fantastic elements, and loses its cathartic value. For a child and adolescent, a traumatic event is any event that overrides their ability to manage and regulate emotional reactions. The ability to handle such experiences depends on the stage of brain development. In an attempt to find an adaptive solution to trauma, subjects at a developmental age exposed to ACEs manifest behaviors to manage post-traumatic suffering, an attempt that often becomes ineffective and dysfunctional, as clearly emerged in the play that our subjects performed.

## 6. Future Research

A peculiarity of the MCAST interview is that the child is repeatedly engaged, both emotionally and cognitively, in an imagined situation that is highly stressful from the point of view of attachment. In children exposed to “Childhood Unfavorable Experiences”, the “state of mind” predominating over the representation of attachment is, therefore, disorganized (D) and insecure (A–C), and prognostically negative. The MCAST presents itself as a valid tool both from a psychodiagnostic point of view for the specificity with which it identifies the attachment pattern. From the psychotherapeutic point of view, when they are configured as “triggers” of traumatic experiences actually experienced, they can become therapy's targets for the treatment of ACEs (such as EMDR therapy).

It is interesting to investigate whether the use of play in the clinical setting could be considered the border between a psychodiagnosis and psychotherapy. In fact, in the first phase of evaluation, which represents the first step of the therapy, this use could be aimed at better understanding the child’s post-traumatic functioning (e.g., dissociative, hyperactive, and/or depressive type) and setting the most appropriate and personalized therapeutic plan. The psychotherapy should, therefore, focus on processing the experiences absent in the child’s personal narrative due to fragmented traumatic memories not present in the autobiographical memory. Such traumatic memories are sensitive to stimuli of new experiences, leaving the traumatized child in a perpetual state of hyperarousal. This processing process can occur naturally through play. Children cope with trauma and related stress by playing in similar situations and learning to cope with it gradually. The role of the therapist, in many cases, would be to establish the proper conditions for this process to happen on its own; at other times the therapist can guide the child through structured or semi-structured play and encourage the child to relive the process event in a different way, and with a more adaptive result than the original event.

## 7. Limitations 

The present study shows some limitations. First of all, the use of the MCAST, which involves the observation of “exploratory play”, assumes that a satisfactory resolution of the activity can lead to different exploratory behavioral patterns, characterized by a relaxed, imaginative, progressive, and pleasurable quality (neither hyperarousal nor hyperarousal), linked to imagination and a sense of mastery [33]. This was not observed in our clinical sample of victimized children so, this variable, as can happen in research, was not coded. Furthermore, it would have been possible to use the ASCT version [39], which contemplates a recent classification method, such as Miljkovitch and Pierrehumbert’s CCH [40].

## Figures and Tables

**Table 1 children-09-01991-t001:** Types of ACEs in the target group.

Attachment Style Type	N° Subjects (n° 17)
Sexual abuse	4
Physical maltreatment Psychological maltreatment and neglect Mental illness of the parent	4 2 5 2

**Table 2 children-09-01991-t002:** Attachment style frequency in the target group.

Attachment Style Type	N° Subjects (n° 17)
Secure	2
Avoidant Anxious-Ambivalent Disorganized-Disoriented	2 3 10

**Table 3 children-09-01991-t003:** Mean frequencies of post-traumatic child's play (PTCP) markers.

PTCP Markers	Mean Frequencies of the Responses to the Four Core MCAST Stories (n° 17)
Repetition Revisiting Hyperarousal Dangerousness Happy ending	12 11.5 14.75 12.5 7

## Data Availability

Written informed consent was obtained from the subject(s) in order to publish this paper.

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
