# Peer review of "Post-Traumatic Play in Child Victims of Adverse Childhood Experiences: A Pilot Study with the MCAST—Manchester Child Attachment Story Task and the Coding of PTCP Markers"

_children, 2022, doi:10.3390/children9121991_

Round 1

Reviewer 1 Report

This is a very interesting topic that has been little addressed by scientific research due to its costly time and dedication, with abundant theoretical support (too much), so I believe it would be good to publish with some major changes.

Perhaps the title is misleading and I would change the word "study" for "pilot study" given the number of cases that have been explored (N= 17).

The theoretical introduction is too long and should be shortened. Perhaps the problem is that the markers of Terr's post-traumatic play are presented in this section and if I understand the proposal correctly, it is these same markers that are evaluated, and therefore, they should appear in the methodology section, made explicit as the instrument that appears in the title (PTCP makers).

This methodology section should also explain the procedure for measuring these markers and whether there has been an inter-coder evaluation.

Perhaps it would also be good to explain that the MCAST is a version of the ASCT (Bretherton, Ridgeway, & Cassidy, 1990), as well as that there are widely used classification methods such as the CCH by Pierrehumbert i collaborators (Miljkovitch & Pierrehumbert, 2008).

The other key point that needs further revision is the results section. In this section it is expected to find the descriptive statistics of all the tests that have been passed (for example: CBCL, CDI, TSCC, CSBI...).

It would also be convenient to show the results of the total ACEs (i.e., answering the question How many children with 0 ACEs, how many with 1 ACE, how many with 2 ACEs, etc.? Apart from the description of the ACE typologies.

The description of how the MCAST narratives have been analyzed (lines 242 to 266) should be moved to Methodology.

The PCTCs in Table 2 should be explained how they are scored in the methodology section (I was pointing out earlier that perhaps it is the Terr markers, but in that case the markers with 0 frequency should also appear in the results table).

The discussion section can also be improved, reducing the theoretical reflections (some redundant and not related to the results of this research for example) and attending to the following topics (as a proposal):

- Interest of the present research

- Interest of the results and of the importance of taking into account both ACEs and indicators of Trauma in play

- Weaknesses of the research presented

- Proposals for future research.

Proposed minor changes:

Line 35 - Would propose to change "natural language" to "form of communication".

Line 55 - I would propose to change "chronological" to "narrative".

I wish the authors all the very best with this study.

References;

Bretherton, I., Ridgeway, D. and Cassidy, J. (1990) Assessing Internal Working Models of the Attachment Relationship. In: Greenberg, M.T., Cicchetti, D. and Cummings, E.M., Eds., Attachment in the Preschool Years: Theory, Research, and Intervention, University of Chicago Press, Chicago, 273-308.

Miljkovitch, R., & Pierrehumbert, B. (2008). Des stratégies comportementales d’attachement aux stratégies représentationnelle: construction et validité des cartes de codage pour les 

Author Response

This is a very interesting topic that has been little addressed by scientific research due to its costly time and dedication, with abundant theoretical support (too much), so I believe it would be good to publish with some major changes.

Authors: The authors thank them for their feedback and valuable suggestions.

Perhaps the title is misleading and I would change the word "study" for "pilot study" given the number of cases that have been explored (N= 17).

Authors: changed "study" to "pilot study"

The theoretical introduction is too long and should be shortened. Perhaps the problem is that the markers of Terr's post-traumatic play are presented in this section and if I understand the proposal correctly, it is these same markers that are evaluated, and therefore, they should appear in the methodology section, made explicit as the instrument that appears in the title (PTCP makers).

Authors: we have reshaped the Introduction making it shorter and clearer and we have moved the specifications of the PTCP markers to the methodological section.

This methodology section should also explain the procedure for measuring these markers and whether there has been an inter-coder evaluation.

Authors: We have further clarified what type of Inter-coder evaluation was used

Perhaps it would also be good to explain that the MCAST is a version of the ASCT (Bretherton, Ridgeway, & Cassidy, 1990), as well as that there are widely used classification methods such as the CCH by Pierrehumbert i collaborators (Miljkovitch & Pierrehumbert, 2008).

Authors: We have clarified and specified this point in the "Limitations" section.

The other key point that needs further revision is the results section. In this section it is expected to find the descriptive statistics of all the tests that have been passed (for example: CBCL, CDI, TSCC, CSBI...).

Authors: We made it clear in the objectives, as well as in the methodological section, that this study is part of a much larger research-intervention project and that in order to deepen the specific clinical objectives revealed in the assessment and observation, we focused on the results of the MCAST.

It would also be convenient to show the results of the total ACEs (i.e., answering the question How many children with 0 ACEs, how many with 1 ACE, how many with 2 ACEs, etc.? Apart from the description of the ACE typologies.

Authors: We have included a specific table.

The description of how the MCAST narratives have been analyzed (lines 242 to 266) should be moved to Methodology.

Authors: Done

The PCTCs in Table 2 should be explained how they are scored in the methodology section (I was pointing out earlier that perhaps it is the Terr markers, but in that case the markers with 0 frequency should also appear in the results table).

Authors: We have improved the explanation of the table and the evaluation and coding of markers.

The discussion section can also be improved, reducing the theoretical reflections (some redundant and not related to the results of this research for example) and attending to the following topics (as a proposal):

- Interest of the present research

- Interest of the results and of the importance of taking into account both ACEs and indicators of Trauma in play

- Weaknesses of the research presented

- Proposals for future research.

Authors: We have improved the 'Discussion' section by removing narrative and redundant statements. We have included 'Limitations' and 'Future Research' paragraphs.

Proposed minor changes:

Line 35 - Would propose to change "natural language" to "form of communication".

Authors: Done

Line 55 - I would propose to change "chronological" to "narrative".

Authors: Done

I wish the authors all the very best with this study.

Authors: Thank you for your good wishes and for taking the time to contribute!

References;

Bretherton, I., Ridgeway, D. and Cassidy, J. (1990) Assessing Internal Working Models of the Attachment Relationship. In: Greenberg, M.T., Cicchetti, D. and Cummings, E.M., Eds., Attachment in the Preschool Years: Theory, Research, and Intervention, University of Chicago Press, Chicago, 273-308.

Miljkovitch, R., & Pierrehumbert, B. (2008). Des stratégies comportementales d’attachement aux stratégies représentationnelle: construction et validité des cartes de codage pour les 

Reviewer 2 Report

I found the study very interesting and relevant to therapists working in the tradition of play therapy and attachment responses. As a researcher working with adolescents, this was useful to think about the impact of ACE's on later development and play. It would be a real strength to the paper to discuss how the impact of these findings could impact on the life course and future outcomes, but I am aware this may be another paper.

Few minor corrections to consider-

Pge 2 line 47- While deprived environments can impact negatively on development outcomes, it is not classified as an ACE

Pge 4 line 156 spelling error- someone change to somewhat

Pge 5 line 236- this whole number is 15 children as opposed to 17 as reported earlier

Author Response

I found the study very interesting and relevant to therapists working in the tradition of play therapy and attachment responses. As a researcher working with adolescents, this was useful to think about the impact of ACE's on later development and play. It would be a real strength to the paper to discuss how the impact of these findings could impact on the life course and future outcomes, but I am aware this may be another paper.

Authors: Thank you for your appreciation and feedback which has certainly improved our study!!!

Few minor corrections to consider-

Pge 2 line 47- While deprived environments can impact negatively on development outcomes, it is not classified as an ACE

Authors: we have corrected it

Pge 4 line 156 spelling error- someone change to somewhat

Authors: Done

Pge 5 line 236- this whole number is 15 children as opposed to 17 as reported earlier

Authors: it was a typo. We apologise and have corrected it.

Round 2

Reviewer 1 Report

Dear authors,

The article has, in my opinion, undergone significant improvement, even though there are still some crucial areas that require improvement.

As the 11 play characteristics in Lenore C. Terr's original article do not exactly match the 10 proposed in the present paper, this should be mentioned. For example, In line 90 you could put something like: "For the purpose of XXXX the authors of this paper have reformulated some of the 11 characteristics proposed by Lenore C. Terr (1981) into these 5 characteristics of post-traumatic play". So that this could be more clearly related to Table 3 (with the PSTYC Markers).

This would also be facilitated if, in the same description, a title were added:

1. Repetition. Referred also as Compulsive repetitiveness: children who have been traumatized play their "game" repeatedly until they are invited to stop or until they reach an emotional state.

2. Revisiting. the possibility of tracing post-traumatic play to earlier trauma in that it is possible to bring out the traumatic traces of early trauma through play in order to reshape the states of distress related to it.

3. And so on...

If the 11 original characteristics were to be mentioned, they should all be put with the same titles (e.g. the category "Carrying Power of Strenth of the Play to Involve Nontraumatized Youngsters" is not there) etc. And then in Method section you could present the "PSTD grid" as a tool that you have created from Terr's proposals.

In line 90:

"reveal the markers of post-traumatic play (PTCP) that have developed in child victims of ACEs."

And it should read:

"if that reveals any of the 5 post-traumatic play markers (PTCP) proposed in this study (based on Terr (1981))."

In Methodology section it is still not explained how the PTCPs have been categorized. I understand that this is precisely the novel proposal of this study, and as such, there should be some guidelines (or Manual) describing how it was done, style:

In Procedure section  - From the the four core MCAST stories the first author (D.E.) and the second author (L.C.) independently coded the presence/non-presence of the 5 PTCP markers. Scoring agreement was almost 100%, and only on a couple of occasions were differences dissolved by mutual agreement*. The following criteria were used to categorize the MCAST stories (what would become the PTCP grid):

Repetition - to code a story with compulsing repetitiveness the main narrative or plot of the child’s play should be repeated more than twice, with or without different nuances.... etc.

[I'm obviously making this up.]

* This point refers to the need to address the issue of inter-coder reliability.

This point is important if we take into account that as a scientific article the replicability of the article should be very clear. That is to say, the authors should give all the sufficient information so that anyone can replicate their research. If you would retain the copyright of the PSTC grid you must cite the Manual of the codifier, so that any researcher can request it to use it with the conditions they create, but it cannot be hidden.

On the other hand, as it is not mentioned anywhere that the authors have been trained as trainer coders of MSCA by Professor Jonathan Green or collaborators (About MCAST - University of Manchester), this should be indicated as a possible weakness of the study.

In relation to the inter-rater reliability of MCAST, the option created on line 215 is not valid, the one that should be made explicit is the one related to this study. That is, usually one author codes all the participants, and then another expert (co-author or not; and it can be more than one person) codes a second time a part of the same material (usually between 10% or 20%). The two codings are compared to check that the main coder has no biases or errors that would be attributed to this person's way of coding. If the inter-coder coincidence is high, this means good reliability, i.e. the error due to the coder is minimal (or acceptable).

On the other hand, in the Results section, the ACEs should be presented in two forms:

(a) Frequency and percentage of adverse childhood experiences (ACEs). As in https://www.researchgate.net/profile/Esteban-Ezama-Coto/publication/345992829/figure/tbl2/AS:959133767446531@1605686718920/Frequency-and-percentage-of-adverse-childhood-experiences-ACEs_W640.jpg

and,

b) Frequency and percentage of ACEs by type. As in: https://www.researchgate.net/profile/Esteban-Ezama-Coto/publication/345992829/figure/tbl3/AS:959133767458818@1605686718951/Frequency-and-percentage-of-ACEs-by-type_W640.jpg

In relation to this, it is strange that ACEs are not repeated in the same subject (for example, children who present physical abuse also tend to suffer from neglect). The interesting thing about using the measure proposed by Felitti is precisely to take into account that it is the accumulation of ACEs that has the clearest impact as a factor in itself on the mental and physical health of individuals.

In this same RESULTS section, there is a lack of a part where the three variables on which the work is focused are related (ACE, APEGO-MCAST- and PTCP). If you do not want to use any statistics given the infrequency of the data, first at all, you can delete the use of any Software Statistical Package (SPSS) in Method Section, and you could set up a descriptive table with the information you believe is relevant. You could take as an example the one presented by Lenore C. Terr in her article (1981) -although with the data related to ACE, ATTACHMENT evaluated from the method proposed by MCAST and PTCP).

Lines 284 to 305 should go to PROCEDURE, in the (missing) section that describes how an MCAST history has been evaluated from the PTCP grid.

In the Discussion section, line 313, the specification “with reference to the age of our group,” is not understood and should be clarified or deleted.

The addition “Furthermore, it would have been possible to use the ASCT version [39], which contemplates a recent classification method such as Miljkovitch & Pierrehumbert's CCH [40].” It should also be deleted or modified.

Apart from the proposals that I offered, there are also others (Hodges, Steele, Hillman, & Henderson, 2003, Bretherton & Oppenheim, 2003). Perhaps in the same title it should be indicated that the study was carried out with an "attachment-based stories stem" or "Narrative Story Stem Methodology" (which is common in all the commented systems), and then in methodology explain that MCAST has been used in this study to code the attachment of participants. I am commenting this because I think that the contribution you make is not the relationship of the ACE with the MCAST and the PTCP markers, but rather that your contribution is the PTCP markers that could eventually be applied in any attachment classification system based on attachment-based story steems. This would be my impression of your work, which I repeat, I find very interesting, but with an unclear structure (in the context of a scientific publication).

Maybe: Post-Traumatic Play in Child victims of Adverse Childhood Experiences: a pilot study with PTCP markers of the Narrative Story Completion Task 

Author Response

Dear Reviewer,
 we would like to bring to your attention an issue that could distort the nature of our study.
In the first round, we responded extensively to the valuable your suggestions, which certainly improved our paper; but at this late stage, we think there is a misunderstanding in your  revisions.
It is implicit that we are neither standardising nor validating the MCAST so the claims about coding specifications are simply a belittling of the work done by Prof. Green and collaborators, which is widely cited in both methodology and results.
Furthermore, we are already aware that to use MCAST in research there is a need for training with Prof. Green, which of course the author D.D. has done, otherwise we could not ethically propose the research.
To remove the acronym MCAST from the title and freely replace it with another tool (not used by us) seems to go against the objectives of our research study.
You  have used tools that do not correspond to those described in our study and that therefore the suggested changes are impossible for us to make.

We apologize, but we are unable to proceed with the revisions you indicated
to us in the second round. Thanks for your thorough work